# Circulating Peptidome Is Strongly Altered in COVID-19 Patients

**DOI:** 10.3390/ijerph20021564

**Published:** 2023-01-14

**Authors:** Gianluca Baldanzi, Beatrice Purghè, Beatrice Ragnoli, Pier Paolo Sainaghi, Roberta Rolla, Annalisa Chiocchetti, Marcello Manfredi, Mario Malerba

**Affiliations:** 1Department of Translational Medicine, University of Piemonte Orientale, 28100 Novara, Italy; 2Center for Translational Research on Autoimmune and Allergic Diseases, University of Piemonte Orientale, 28100 Novara, Italy; 3Respiratory Unit, Sant’Andrea Hospital, 13100 Vercelli, Italy; 4Internal and Emergency Medicine Department, Department of Translational Medicine, University of Piemonte Orientale, 28100 Novara, Italy; 5Department of Health Sciences, University of Piemonte Orientale, 28100 Novara, Italy

**Keywords:** COVID-19, peptidomics, protein degradation, biomarkers, respiratory disease

## Abstract

Whilst the impact of coronavirus disease 2019 (COVID-19) on the host proteome, metabolome, and lipidome has been largely investigated in different bio-fluids, to date, the circulating peptidome remains unexplored. Thus, the present study aimed to apply an untargeted peptidomic approach to provide insight into alterations of circulating peptides in the development and severity of SARS-CoV-2 infection. The circulating peptidome from COVID-19 severe and mildly symptomatic patients and negative controls was characterized using LC-MS/MS analysis for identification and quantification purposes. Database search and statistical analysis allowed a complete characterization of the plasma peptidome and the detection of the most significant modulated peptides that were impacted by the infection. Our results highlighted not only that peptide abundance inversely correlates with disease severity, but also the involvement of biomolecules belonging to inflammatory, immune-response, and coagulation proteins/processes. Moreover, our data suggested a possible involvement of changes in protein degradation patterns. In the present research, for the first time, the untargeted peptidomic approach enabled the identification of circulating peptides potentially playing a crucial role in the progression of COVID-19.

## 1. Introduction

Coronavirus disease 2019 (COVID-19) is an infectious and highly contagious respiratory disease caused by the severe acute respiratory syndrome coronavirus 2 virus (SARS-CoV-2). To date, more than 312 million COVID-19 confirmed cases and over 5 million deaths have been reported globally [1]. It is well known that SARS-CoV-2 infected patients’ can be asymptomatic, or they can manifest mild non-specific symptoms, including fever, dry cough, headache, and gastrointestinal distress, but they could also present severe pneumonia, including organ function damage. In addition, the presence of comorbidities (e.g., cardiovascular diseases, immunosuppression, diabetes) could worsen the COVID-19 prognosis [2].

It is precisely this complex clinical spectrum and the need for personalized medical care that have led the scientific community to focus on the discovery of a diagnostic and prognostic signature of COVID-19 through multi-omics approaches [3]. Proteomic workflow has been applied to the elucidation of the interaction between the COVID-19 virus and host proteins by mainly using cross-linking and liquid chromatography-tandem mass spectrometry (LC-MS/MS) techniques [4,5,6,7,8]. Equally important, however, have been the characterization of the proteomic profile of host cells infected by SARS-CoV-2 and the comprehension of the virus-driven alterations of metabolic homeostasis in positive patients [6,9,10,11,12,13,14]. Metabolomics, as well, helped get an insight into the pathophysiological alterations related to SARS-CoV-2 infection, since changes in the metabolomic profile have been deepened both in severe and mild infected cohorts, aiming to predict the disease course.

Although to date many analyses of biofluids such as plasma, serum, exhaled breath condensate, urine, and stool samples have been performed in the omics field [2,15,16,17,18], including a wide range of proteomic, metabolomic, lipidomic, and genomic studies [19,20,21,22,23], nothing is known about the role of circulating peptides during diseases. Small bioactive peptides, which can be easily found in cells and biological fluids, play a crucial role in the maintenance of physiological functions: any changes in their concentration and structure might be indicative of pathological conditions. Moreover, since low molecular weight (LMW) peptides are smaller than proteins, they can passively diffuse across membranes and endothelial barriers [24]. Among the most important peptides, there are the human antimicrobial peptides (AMPs), predominantly cationic, and amphiphilic α-helical peptides, which are considered an intrinsic part of the innate immune system and a defense strategy against a broad spectrum of bacteria, fungi, viruses, and parasites. Particularly, antiviral peptides (AVPs, 8–40 amino acids in length), such as RTD-1 and HD5 exploit their antimicrobial activity through different mechanisms of action. Extracellularly, they can provoke membrane lysis on enveloped viruses or prevent the interaction between virus and host cells. Intracellularly, they could interact with the intracellular target, thus blocking viral replication at various steps, and avoid the integration of the viral genetic load into human DNA by inhibiting reverse transcriptase and viral proteases [25].

Peptidomic analysis is an emerging technique that can be used for the identification of endogenous or exogenous peptides related to pathogenic organisms and infectious agents. Peptidomic studies, which have the advantage of not requiring chemical or enzymatic digestion for sample processing, can provide insight into the specific proteolytic activities associated with a pathophysiological context [26].

On the other hand, peptidomics has also several disadvantages that have limited its application in the clinical setting. From an analytical point of view, peptides have a wide range of polarity and concentration that makes their detection very difficult, unless a specific protocol for peptide enrichment and clean-up is applied. Furthermore, highly abundant proteins might have a masking effect on less abundant peptides, and the bioinformatic identification is difficult as the site of enzymatic digestion cannot be specified [27,28].

Blood (plasma and serum) samples are of particular interest in the clinical setting for discriminating between healthy and disease states. Circulating peptides can provide a wide range of diagnostic and biological information based on the length and aminoacidic sequence of the peptide, the identity of the originating protein, the peptide’s abundance, and the nature of the carrier protein to which the peptide could be bound [29]. Among the different body fluids, plasma is generally the most appropriate biological matrix for peptidomic analysis since, contrary to serum, it ensures greater peptide stability. Indeed, the activity of endopeptidases and exopeptidases due to the activation of clotting for serum collection can lead to protein degradation and the consequent release of new peptides [30,31].

The objective of the present study was to decipher the role of circulating peptides during SARS-CoV-2 infection. A peptidomic workflow was chosen over a proteomic one because, to our knowledge, to date, the circulating plasma peptidome associated with COVID-19 has not been yet investigated. In addition, this technique explores naturally occurring and endogenous LMW peptides and proteolytic fragments, which are considered pathophysiological surrogates in signaling, proteolytic, and anti-proteolytic pathways in many systemic diseases [32].

Plasma samples from patients with COVID-19 were analyzed through a mass-spectrometry-based peptidomic approach to map the circulating peptidome and to identify peptide markers potentially reflecting the molecular pathophysiology of disease progression.

## 2. Materials and Methods

### 2.1. Patients

Plasma citrate samples from 21 subjects who tested positive for SARS-CoV-2 and were admitted from April 2020 to April 2021 to the Sant ‘Andrea Hospital of Vercelli and the Maggiore Della Carità Hospital of Novara for pneumonia and/or respiratory failure were collected. COVID-19 patients were subdivided into two groups depending on disease severity, namely, severe (*n* = 10) and mildly symptomatic (*n* = 11) subjects. We defined severe patients as those admitted to the semi-intensive care respiratory unit with respiratory failure who required mechanical ventilation (continuous positive airway pressure, CPAP), while mildly symptomatic subjects were those with mild to severe respiratory failure requiring only non-invasive oxygen supplementation. The clinical characteristics and comorbidities of the patients are reported in Table 1. Healthy individuals were enrolled as controls in the hospital environment as well (*n* = 12). In our study, the ‘healthy’ status is determined by negative reverse-transcriptase polymerase chain reaction (RT-PCR) for SARS-CoV-2 and by the absence of other comorbidities.

The Institutional Review Board (Comitato Etico Interaziendale Novara) approved this study (n. RQ06320/25 March 2020).

### 2.2. Peptide Extraction

Citrate tubes were used to obtain plasma citrate samples from all experimental groups. Citrate tubes were centrifuged for 15 min at 1600 rpm at room temperature. The obtained plasma was thus aliquoted and stored at −80 °C until use. Peptides from plasma samples were extracted using a QAE Sephadex A-25 strong anion exchange resin (Sigma-Aldrich, St. Louis, MO, USA). Briefly, 160 µL of sorbent was washed twice through the addition of 400 µL of buffer (0.02 M Tris, pH = 8.26) and subsequently centrifuged at 800× *g* for 10 s. The supernatant was then carefully removed. Before extraction, 200 µL of plasma was diluted with 400 µL of buffer and then added to the washed resin. After a 30 min shaking incubation (room temperature, 1000 rpm), the samples were centrifuged at 500× *g* for 10 s and the supernatant was removed. The sorbent was again washed thrice with 700 µL of buffer (500× *g*, 10 s) and 800 µL of 0.5 % formic acid (elution buffer) was added. Subsequently, samples were incubated for 15 min while vortexing.

For the desorption of peptides from the surface of highly abundant proteins, the eluates deriving from the fractionation on anion exchange sorbent were incubated at 60 °C for 15 min [33]. After heating, samples were desalted on the Discovery^®^ DSC-18 solid-phase extraction (SPE) 96-well plate (25 mg/well, Sigma-Aldrich Inc., St. Louis, MO, USA). The SPE plate was preconditioned with 1 mL of acetonitrile and 2 mL of water. Following sample loading, the SPE was washed with 1 mL of water and the adsorbed peptides were eluted with 800 μL of acetonitrile/water (80:20). Samples were vacuum-evaporated and reconstituted in 11 µL mobile phase (0.1% formic acid in water) for the analysis.

### 2.3. LC-MS/MS Analyses

The circulating peptidome was characterized using the micro-LC Eksigent Technologies (Eksigent, Dublin, CA, USA) system including a micro LC200 Eksigent pump with flow module 5–50 µL and interfaced with a TripleTOF 5600+ mass spectrometer (AB Sciex, Concord, Canada) equipped with DuoSpray Ion Source and Calibrant Delivery System. The stationary phase was a Halo C18 column (0.5 × 100 mm, 2.7 μm; Eksigent Technologies Dublin, CA, USA). The mobile phase was a mixture of 0.1% (*v*/*v*) formic acid in water (A) and 0.1% (*v*/*v*) formic acid in acetonitrile (B), eluting at a flow rate of 15.0 μL/min with an increasing concentration of solvent B from 2% to 40% in 30 min. Injection volume was 6.0 μL and oven temperature was set at 40 °C.

For identification and quantification purposes the mass spectrometer analysis was performed using a mass range of 100–1500 Da (TOF scan with an accumulation time of 0.25 s), followed by an MS/MS product ion scan from 200 to 1250 Da (accumulation time of 5.0 ms) with the abundance threshold set at 30 cps (35 candidate ions can be monitored during every cycle). The ion source parameters in electrospray positive mode were set as follows: curtain gas (N2) at 25 psig, nebulizer gas GAS1 at 25 psig, and GAS2 at 20 psig, ion spray voltage floating (ISVF) at 5000 V, source temperature at 450 °C, and declustering potential at 25 V. The MS data were acquired with Analyst TF 1.7 (ABSCIEX, Concord, Canada).

### 2.4. Peptide Database Search

The MS files were searched using Protein Pilot software v. 4.2 (AB SCIEX, Concord, Canada) and Mascot v. 2.4 (Matrix Science Inc., Boston, MA, USA). For Mascot, no enzyme was used and the following modification for the assay was specified: oxidized methionine as variable modification. An assay tolerance of 50 ppm was specified for peptide mass tolerance, and 0.1 Da for MS/MS tolerance. The peptide charges to be detected were set to 2+, 3+, and 4+, and the assay was set on monoisotopic mass. The UniProt Swiss-Prot reviewed database containing human proteins (version 2018.07.07, containing 42,131 sequence entries) was used and a target-decoy database search was performed. False Discovery Rate (FDR) was fixed at 1%.

For the peptide matching to the protein of origin and identification of cutting sites, we used PIR (https://research.bioinformatics.udel.edu/peptidematch/, (accessed on 10 November 2022) [34]. For the examination of the protease cleavage site and the searching of putative proteases we used MEROPS—the Peptidase Database (https://www.ebi.ac.uk/merops, (accessed on 10 November 2022) and Proteasix (http://www.proteasix.org/, (accessed on 9 November 2022). For the reference levels of proteins, the covid-omics.app was used [35]. Bioinformatic analysis was also carried out using STRING software v. 11.5 (https://string-db.org, (accessed on 29 May 2022) and Panther software v. 17.0 (www.pantherdb.org, (accessed on 29 May 2022) [36].

The mass spectrometry proteomic data have been deposited to the ProteomeXchange Consortium via the PRIDE partner repository with the dataset identifier PXD036020.

### 2.5. Statistical and Bioinformatic Analysis

Statistical analyses and figures were performed using MetaboAnalyst software v. 4.0 (https://www.metaboanalyst.ca/, (accessed on 20 May 2022) [37]. No data filtering was applied during the processing, and pareto scaling normalization was used to preprocess the data. Peptide abundances were measured as the area under the peak of each peptide using a label-free quantification approach. The final abundances were normalized for the sum of the total area of peptides to reduce the variability caused by a possible different loading of peptides into the column.

## 3. Results

### 3.1. Plasma Peptide Alterations in COVID-19 Patients

Untargeted peptidomic mass spectrometry-based analysis was performed on plasma samples from SARS-CoV-2 patients to investigate whether the circulating peptidome was influenced by the viral infection. An overview of the experimental design is provided in Figure 1. In detail, out of 33 individuals, 21 tested positive for SARS-CoV-2 (COVID-19 group) and 12 were enrolled as negative controls (negative group). None of the individuals enrolled in the study were vaccinated. The clinical characteristics of the patients are reported in Table 1.

Since variable coagulation could contribute to systematic bias and fibrinogen could be connected to sample handling, we standardized as much as possible blood collection and processing. Indeed, blood collection tubes with sodium citrate anticoagulant were employed and samples were immediately centrifuged after collection. The obtained platelet-poor plasma was thus aliquoted and stored at −80 °C until extraction and kept on ice once thawed.

Moreover, considering that peptides can easily undergo degradation in biological matrices, we implemented the peptide extraction method to minimize endogenous proteolytic enzyme activity.

Our untargeted plasma peptidomic approach allowed the identification of a total of 774 different peptides, with only 59 peptides in common among the three groups of samples. Peptide fragmentations of the most relevant peptides are reported in Appendix A. Notably, these results clearly showed a high heterogeneity of identified peptides within the three groups of patients, as shown in the Venn diagram reported in Figure 2a. In addition, the comparison of the peptidome of plasma samples from controls and COVID-19 positive patients highlighted an inverse correlation between the total number of peptides and disease severity. It is difficult to explain why severe patients shared more peptides with negative subjects (170) than with mild patients (66). Peptides are involved in many physiological functions and passively diffuse through cell membranes and epithelial barriers; thus, it is difficult to discriminate peptides generated by the pathological condition and the ones derived from physiological processes. However, the Venn representation does not consider quantitative information, which is always the most important from the biological point of view. In fact, identified peptides within each group may have different abundances or may be not present in all samples. It is also important to remark that these preliminary results were obtained on a small number of patients and that only the statistical analysis performed on peptide abundances can give a clear photograph of biological differences and similarities.

The statistical analysis performed using a *t*-test and the ratio of the abundances of quantified peptides within each group (*p*-value ≤ 0.05, fold change ≥ 1.3), revealed the presence of 73 modulated peptides in COVID-19 patients compared to negative controls. Fifty-eight peptides were regulated in plasma samples from mildly symptomatic COVID-19 patients vs. controls, while 39 peptides were modulated in critical patients. The hierarchical clustering heatmap analysis, shown in Figure 2b, displays the fold changes of the top modulated peptides. The considered heatmap pointed out the three clusters of samples and different peptide levels. The list of most modulated peptides is reported in Table 2, while the complete lists of all modulated peptides are reported in Appendix A.

The main regulated peptides in mildly symptomatic and severe patients compared to controls derives from inflammatory, immune-response, and coagulation proteins.

Interestingly, a recent paper demonstrated that glucocorticoid treatments altered the levels of APOC3, TTHY, ITIH2, HPT, and C1QB. Although our analysis was performed at the diagnosis, before any drug treatment, a modulation of peptides related to these proteins was detected in our dataset, suggesting their potential role in the resolution of the disease [38]. 

### 3.2. The Circulating Peptidome Reflects the Disease State

To assess the overall differences between COVID-19 severe and mildly symptomatic patients and controls, peptide abundances were further analyzed using multivariate statistical analysis. Supervised partial least square discriminant analysis (PLS-DA) was calculated to achieve maximum separation between the three experimental groups. The validity of the PLS-DA model against over-fitting was assessed by the parameters R^2^ (0.87), and the predictive ability was described by Q^2^ (0.47) (Appendix A). Although the groups are separated enough (Figure 3a), PLS-DA is not able to clearly discriminate the two clusters, suggesting that the features that contribute the most to the separation between the mild and severe patients are unrelated to COVID-19. 

The most predictive or discriminative features that are potentially useful in helping sample classification were also determined through the VIP (variable of importance in projection) score. The VIP score summarized the most prominent peptides contributing to the observed phenotypic variations in the COVID-19 plasma samples (Figure 3b).

Peptidomic differences between the three patient groups were mostly due to peptides derived from fibrinogen alpha chain (FIBA_HUMAN), albumin (ALBU_HUMAN), apolipoprotein A-II (APOA2_HUMAN), and haptoglobin (HPT_HUMAN).

The bioinformatic analysis performed on modulated peptides (Figure 4) highlighted their derivation by proteins connected in a few functional networks: such as complement and coagulation cascades (namely CFB, C4A, C3, C1QB, SERPIN, FGA, and FGB, red dots, Figure 4a) and activation of C3 and C5 (violet dots, Figure 4a). In addition, the classification of peptides based on the function of the originating protein revealed a higher presence of peptides linked to protein binding-activity modulators (24%), scaffold/adaptor proteins (24%), and defense/immunity proteins (19%) in line with an infectious disease (Figure 4b).

Finally, we used the Proteasix peptide-centric prediction tool to identify the putative proteases that may be responsible for the generation of peptides changing between COVID-19 and negative samples. This hypothesis is based on the cleavage site sequence and points to members of the metalloproteinase family, cathepsin family, meprin A, pepsin A-3, and calpain1/2 as the proteases putatively generating the regulated peptides, with no clear differences between upregulated and downregulated ones (Appendix A).

## 4. Discussion

The comparison between the peptidome profile of SARS-CoV-2 positive and negative patients reported the quantitative modulation of 73 peptides from 31 different proteins, the majority of which were downregulated in COVID-19 infected patients (fold change < 0.5). Of those peptides, 14 were from albumin and their decrease is in line with the well-documented decrease in albumin levels in COVID-19 patients [39]. In addition, the decrease of apolipoprotein A-1 derived peptides (10 peptides) is in accordance with the reported decrease of circulating HDL [40]. Furthermore, for apolipoprotein C-III the total levels were reported to be dysregulated in relation to COVID-19, confirming the observed changes in the degradation pattern with an increase of cutting efficiency at position 28 that justifies the increase of S_21_EAEDASL_28_ peptide, and the concomitant decrease of 21–84 and 21–34 fragments upon infection [41,42]. Despite the well-known correlation between apolipoprotein levels and metabolic disorders, the design of the present study did not allow it. In fact, among patients involved in the research, just a few reported metabolic disorders, thus limiting the possibility of identifying potential correlations between apolipoproteins and metabolic diseases. A further study designed properly could contribute to the understanding of this potential relationship. 

Other previous studies performed on healthy subjects identified a larger number of peptides, but these were mainly limited to collagen type I alpha 1 chain, fibrinogen alpha chain, alpha-1 antitrypsin, and thymosin β-4 proteins. These differences are mainly due to the different isolation and analytical methods that were employed by the authors [43,44]. Indeed, we took advantage of a micro liquid chromatography system for our analysis, certainly less sensitive compared to other analytical techniques (nanoLC system, CE-MS), but more reproducible and able to guarantee a high throughput approach. Moreover, our analysis partially overlapped the results obtained in previous works; in addition to peptides deriving from proteins commonly found in plasma samples (transport proteins, enzyme inhibitors, complement factors, etc.), we were able to identify some unique and less abundant peptides (Retinol-binding protein, Plasminogen, Immunoglobulin gamma-1 heavy chain).

Moreover, in accordance with our analysis, a number of peptides deriving from proteins associated with the inflammatory process are decreased in SARS-CoV-2 infected patients: for example, human alpha-1-antitrypsin (A1AT_HUMAN), human inter-alpha-trypsin inhibitor heavy chain H2 (ITIH2_HUMAN), human complement C3 (CO3_HUMAN), and human complement C1q subcomponent subunit B (C1QB_HUMAN) peptides.

Surprisingly, although alpha-1-antitrypsin and inter-alpha-trypsin inhibitors are reported to increase during COVID-19 (20), the derived peptides are decreasing in our analysis (5 concordant peptides for both). This may indicate enhanced stability or escape from degradation may contribute to the increase of circulating levels.

Human alpha-1-antitrypsin (A1AT) is a circulating protease inhibitor mainly synthesized by hepatocytes and targeting neutrophil elastase, whose activity is critical for the prevention of proteolytic tissue (lung and liver) damage [45]. It has been enlightened the dual role of A1AT as an antiviral and anti-inflammation protein. Indeed, alpha-1-antitrypsin not only can protect the lung from damages such as emphysema by inhibiting neutrophilic elastase (anti-inflammation role) but can also block SARS-CoV-2 infection through the inhibition of a protease involved in the entry of SARS-CoV-2 into the host cells (transmembrane protease serine-type 2-TMPRSS2), thus exerting an antiviral action [46,47]. Immune dysregulation and the more severe symptoms shown by COVID-19 patients may be so associated with alterations in A1AT levels.

In the same way, inter-alpha-trypsin inhibitors are an acute phase protein family with matrix protective activity through protease inhibitory action. Since previous studies pointed out that low levels of inter-alpha-trypsin inhibitor proteins correlated with severe sepsis and influenced patient survival [48,49], these observations may help explain the downregulation of human inter-alpha-trypsin inhibitor heavy chain H2 peptide in SARS-CoV-2 infected subjects, when compared to negative controls.

Interestingly, the complement system fragments were under-expressed in positive subjects. As diffuse complement activation is one of the key features of COVID-19, a variation of complement factors derived peptides is not surprising. We observed complement C1q and C3 fragments (respectively, 1 and 6 peptides) decreasing during infection, and C4 changing digestion pattern with 1 peptide downregulated and 3 upregulated. Even if the involvement of the humoral immune response in COVID-19 development has been largely demonstrated, the role of various complement system components in the prognosis of SARS-CoV-2 infection remains nevertheless unclear [50]. According to Wu et al. [51], who investigated the clinical and immunological characteristics of COVID-19 patients stratified on the basis of disease severity, complement C1q subcomponent subunit B levels were, for instance, significantly reduced in severe cases. Conversely, complement C3 levels have been either associated with poor prognosis in severe patients [52,53] or addressed as not predictive of disease progression [54]. It can be assumed that the alterations in the degradation pattern of the considered proteins correlate with the hyperactivation of the complement pathway already demonstrated in SARS-CoV-2 infection.

In contrast, the comparison of COVID-19 positive subjects vs. negative controls also established the upregulation of only a few peptides: more specifically, transthyretin (TTHY_HUMAN), isoform 2 of Haptoglobin (HPT_HUMAN), Ceruloplasmin (CERU_HUMAN), isoform 2 of Fibrinogen alpha chain (FIBA_HUMAN), and isoform 2 of Complement C4-A (CO4A_HUMAN) peptides.

Transthyretin is an acute phase-reactant acting as a hormone transporter whose levels negatively correlate with inflammation, so much that low concentrations of TTHY are indicative of a systemic inflammatory state [55]. Our analysis, in keeping with what we have just reported, highlights a decrease in transthyretin protein levels; our data suggest a parallel change in degradation with an increase cutting at position 137 indicated by increased S_137_TTAVVTNPKE_147_ and the concomitant decrease of 129–147, 130–147, and 116–147 fragments. Although transthyretin protein has been proposed as a marker of nutritional status, the potential contribution of the feeding status in the modulation of identified peptides was excluded [56]. In fact, TTHY is a marker of malnutrition and of prognosis associated with malnutrition, and, although dietary questionnaires were not available, no malnourished patients were reported in the clinical data. In addition, all the samples were obtained at the diagnosis of the infection, as soon as the patients were recovered at the hospital, thus excluding any impact of the therapy and the disease on nutrition. However, further studies are needed to definitely exclude the contribution of feeding status. 

Severe COVID-19 patients often exhibit signs of hemolysis and consequently elevated levels of free heme, whose excess promotes oxidative and inflammatory stress. The acute phase protein haptoglobin (Hpt) can scavenge free heme, preventing heme-induced inflammation, but in the case of severe hemolysis, this protein may be overwhelmed, hence not completely neutralizing heme’s dangerous effects [57]. The observed upregulation of HPT_HUMAN peptide in mild to severe SARS-CoV-2 patients, therefore, led us to postulate that infected subjects show high levels of the related protein, which is unable to efficiently neutralize free heme. Moreover, the presence of an upregulated peptide together with two downregulated ones strongly suggests a change in the degradation pattern.

Similarly, ceruloplasmin (Cp), positive acute phase reactant, and the coagulation factor fibrinogen are known key molecular players in inflammation [58,59]. For this reason, it is not surprising to notice an increase in the corresponding peptide levels and of the whole proteins as well in COVID-19 positive patients.

Lastly, the modulation of complement C4-A peptide in SARS-CoV-2 infected subjects is consistent with the changes in complement pathway activation stated before.

The lists of severe to negative and mild to negative peptides significantly overlap with the previous data, with the interesting exception of a peptide derived from the cytoskeletal protein talin. This peptide is specific for severe patients and one of the features discriminating mild to severe ones. Other differences are instead quantitative, with a stronger upregulation of transthyretin, modulation of haptoglobin peptides, and downregulation of fibrinogen, complement, and alpha-1-antitrypsin in severe versus mild cases.

The broad spectrum of the COVID-19 symptoms seems to be attributable to a variety of pathophysiological mechanisms. Among them, matrix metalloproteinases (MMPs), a family of zinc-dependent and Ca^2+^-containing endoproteinases deriving from inflammatory and parenchymal cells in the lung, have been already reported as involved in pulmonary pathologies characterized by tissue remodeling and acute lung injury (e.g., asthma, COPD) [60].

Despite the involvement of protease activity in COVID-19, it is not easy to link peptide abundance to changes in the activity of specific proteases. A search in the MEROPS protease database [61] indicates that the proteases responsible for the generation of the above-mentioned peptides are largely unknown. A prediction based on the cleavage sites indicates the metalloproteinase family, catepsin family, meprin A, pepsin A-3, and calpain1/2 as proteases generating the regulated peptides. Changes in protease activity are indeed expected in an inflammatory disease with strong neutrophil and complement activation [62]. In particular, a strong increase in matrix-metalloproteinase 9 (MMP-9), whose correlation with acute lung injury and chronic respiratory diseases is a fact, has been reported in COVID-19 patients and associated with respiratory failure [63]. In the same way, a correlation between MMP-3 serum levels and the severity of COVID-19 pulmonary symptoms has been also assessed [64]. Thus, given the observed alterations in the degradation pattern of the cited proteins and the reported involvement of metallopeptidases in SARS-CoV-2 infection, there may be a link between these two phenomena impacting COVID-19 severity. However, it must be noted that the changes in the circulating peptidome observed are the result of a complex balance between target abundance, protease activity, and clearance rate, all of which could be modulated by COVID-19.

## 5. Conclusions

In the present research, for the first time, the circulating plasma peptidome associated with SARS-CoV-2 infection was investigated. The results showed that peptides’ abundances are correlated not only with the infection but also with the severity of the disease. The untargeted quantitative mapping of those peptides showed their derivation from proteins related to inflammation, coagulation, and immune responses. These three biological pathways are well-known players in COVID-19 disease. In addition, comparing our data with the literature, we found that several peptides did not reflect the originating protein abundances, suggesting the presence of other biochemical mechanisms linked to peptide releases and/or uptake by cells.

The observed change in peptidome composition diverges from those previously reported by other inflammatory conditions such as in septic shock patients where induction of protease activity and an increase of peptide abundance correlate with disease severity [65].

Although to date this is the only study that investigated how SARS-CoV-2 infection affects circulating peptides, a larger cohort of patients, including the asymptomatic ones, would allow the identification of more specific pathways and peptides associated with the development of the disease. A larger study with greater statistical power would also help to overcome high inter-variability between subjects.

Finally, it must be noted that there was the presence of a high inter-variability between subjects of the same group, although all the variations reported and discussed here were statistically significant. Thus, further confirmation of the role of these peptides should be performed.

## Figures and Tables

**Figure 1 ijerph-20-01564-f001:**
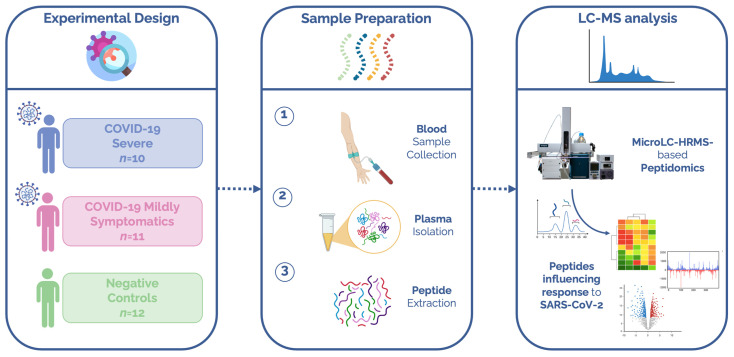
Experimental design of the study. Plasma samples from subjects who tested negative for COVID-19 (*n* = 12) and from COVID-19 positive patients divided into two groups depending on disease severity, namely, severe (*n* = 11) and mildly symptomatic patients (*n* = 12), were used. Circulating peptides were extracted through a strong anion exchange sorbent. Peptides were then characterized using an LC-MS system and the identified and modulated peptides were elaborated with bioinformatics to identify peptides influenced by SARS-CoV-2 infection.

**Figure 2 ijerph-20-01564-f002:**
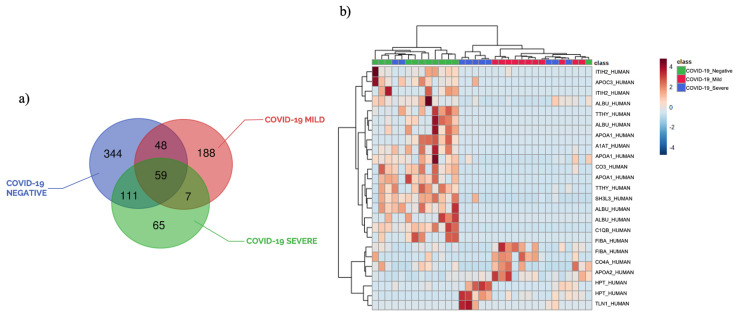
(**a**) Venn Diagram of identified peptides in severe and mildly symptomatic COVID-19 positive patients, and negative controls; (**b**) hierarchical heatmap of identified peptides highlighting the three clusters of samples, with negative controls in green, COVID-19 mild in red, and COVID-19 severe in blue.

**Figure 3 ijerph-20-01564-f003:**
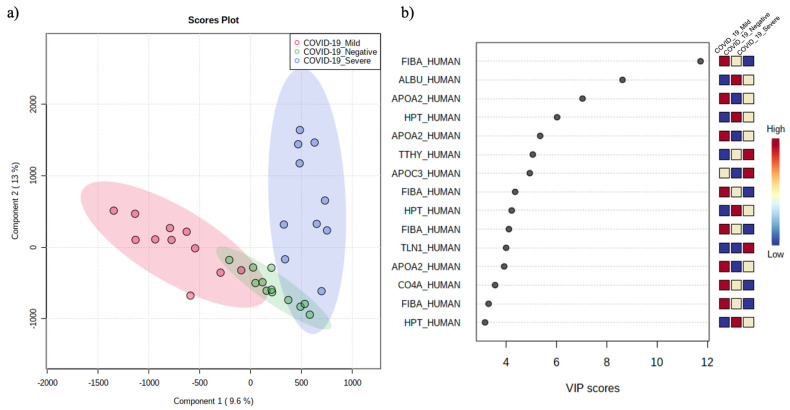
PLS-DA (**a**) and VIP score (**b**) summarizing the most prominent peptides contributing to the observed phenotypic variations in the COVID-19 peptidome.

**Figure 4 ijerph-20-01564-f004:**
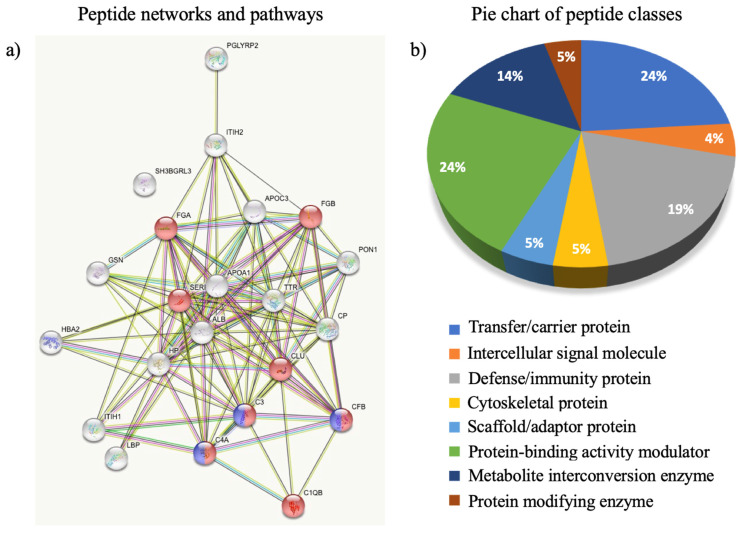
Peptide network and pathways (**a**), complement and coagulation cascades (red dots) and activation of C3 and C5 (violet dots); pie chart of peptide classes (**b**).

**Table 1 ijerph-20-01564-t001:** Clinical data of the patients enrolled in the study.

	Non-COVID-19 Patients	COVID-19 Patients			
	Negative Controls (*n* = 12)	Total (*n* = 21)	Mildly Symptomatic (*n* = 11)	Severe (*n* = 10)	*p*-Value (COVID-19 vs. Negative)
**Demographic Characteristics**					
Female, *n* (%)	6 (50%)	12 (57.1%)	6 (54.5%)	6 (60%)	
Age (years)	75 ± 14.9	74.9 ± 16.1	71.9 ± 21.8	77.2 ± 7.7	0.9813
**Respiratory Support, *n* (%)**					
Continuous Positive Airway Pressure (CPAP)	n/a	10 (47.6%)	0 (0%)	10 (100%)	
Non-Invasive Oxygen Supplementation	n/a	11 (52.4%)	11 (100%)	0 (0%)	
**Dexamethasone Regime, *n* (%)**	n/a	4 (19.04%)	0 (0%)	4 (40%)	
**Outcome, *n* (%)**					
Discharged	12 (100%)	16 (76.2%)	11 (100%)	5 (50%)	
Deceased	0 (0%)	5 (23.8%)	0 (0%)	5(50%)	
**Comorbidity, *n***					
Hypertension	3	11	5	6	
Diabetes	0	4	1	3	
Respiratory System	1	2	1	1	
Cardiovascular System	2	8	4	4	
Other Endocrine System	1	6	3	3	
Chronic Kidney	1	1	1	0	
Digestive System	0	3	0	3	
**Time from Disease Onset to Sample Collection, Days**					
Mean ± SD	n/a	6.45 ± 4.8	5.2 ± 4.7	7.7 ± 3.9	
Range	n/a	1.0–11.0	1.0–11.0	2.0–10.0	

n/a = not applicable.

**Table 2 ijerph-20-01564-t002:** List of most modulated peptides in COVID-19 patients (mild and severe) compared to negative controls with their derived protein, *p*-value, and fold change.

Peptide Sequence	Protein	Modulation	*p*-Value	Fold Change
*VDSGNDVTDIADD*	HPT_HUMAN	Mild/Negative	0.00254	18.8
*SEAEDASL*	APOC3_HUMAN	Mild/Negative	0.00321	6.5
*SEAEDASLLS*	APOC3_HUMAN	Mild/Negative	0.02524	6.4
*DSGEGDFLAEGGGVR*	FIBA_HUMAN	Mild/Negative	0.04475	4.9
*LLSPYSYSTTAVVTNPKE*	TTHY_HUMAN	Mild/Negative	7.96 × 10^−5^	0.0147
*DAHKSEVAHRFKDLGEENFKALVLIAF*	ALBU_HUMAN	Mild/Negative	0.03752	0.0318
*FKVSFLSALEEYTKKLNTQ*	APOA1_HUMAN	Mild/Negative	0.00496	0.0371
*FEIPINGLSE*	ITIH2_HUMAN	Mild/Negative	0.01937	0.0474
*VDSGNDVTDIADD*	HPT_HUMAN	Severe/Negative	0.00186	85.9
*SGASGPENFQVG*	TLN1_HUMAN	Severe/Negative	0.02202	20.0
*TLEIPGNSD*	CO4A_HUMAN	Severe/Negative	0.02502	20.0
*WVQKTIAEN*	HPT_HUMAN	Severe/Negative	0.00546	13.4
*DEAGSEADHEGTHST*	FIBA_HUMAN	Severe/Negative	9.03 × 10^−5^	0.0500
*DDPDAPLQPVTPLQ*	CO4A_HUMAN	Severe/Negative	0.00078	0.0500
*HKSEVAHRFKDLGEENFKALVLIA*	ALBU_HUMAN	Severe/Negative	0.00499	0.0500
*SGFLLFPDMEA*	C1QB_HUMAN	Severe/Negative	0.01355	0.0500

## Data Availability

The mass spectrometry proteomic data have been deposited to the ProteomeXchange Consortium via the PRIDE partner repository with the dataset identifier PXD036020.

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
