# Peer review of "Circulating Peptidome Is Strongly Altered in COVID-19 Patients"

_ijerph, 2023, doi:10.3390/ijerph20021564_

Round 1
Reviewer 1 Report
Circulating peptidome is strongly altered in COVID-19 patients
(IJERPH (ISSN 1660-4601))
General comments
The manuscript presents a novel ‘omics analysis of COVID-19 infection by reviewing the circulating peptidome. All data are made available in a repository (dataset identifier PXD036020), the methodology appears sound, and whilst the population is small (n = 31, insufficient for robust statistical analysis), as a first peptidomic analysis the results should be available to the scientific community.
In general, the standard of written English is good. Some example alterations are provided to allow a better flow or correct minor language issues, but these are suggestions only as the text as written is sufficiently clear.
Comments on content
· Abstract
The Abstract does a good job of highlighting what is novel about the work. You might consider removing the opening sentence completely (any reasonable reader should know what SARS-CoV-2 is by now) and begin instead with “Whilst the impact of coronavirus disease-19 (COVID-19) on the host proteome, metabolome and lipidome in different biofluids has been extensively investigated, to date the circulating peptidome remains unexplored. Thus…"
This change would free up word count capacity to include additional factual description such as your sample size, or to expand on your findings.
· Introduction
Line 38, maybe consider “have been reported globally” instead of “can be counted all over the world”
Lines 44-47, you may wish to consider “Although to date many analyses of biofluids such as plasma, serum, exhaled breath condensate, urine and stool samples have been published …” as a more concise sentence that retains the meaning.
Line 46, Your references are appropriate, but it might be helpful to also include a review paper to capture more of the COVID-19 ‘omics landscape, such as:
https://doi.org/10.3390/ijms23052414
· Materials and Methods
Line 91, it would be helpful to state the dates of recruitment to provide some context as to the likely circulating variant. As it is later stated that no participants were vaccinated, it may well be the case that recruitment took place early in the pandemic, but this should be stated explicitly
Were your healthy controls also recruited in the hospital environment, or somewhere else?
Line 96, was there any difference in medication regime, for example did your work predate the RECOVERY study which led to widespread prescription of dexamethasone to patients receiving CPAP? This is important as glucocorticoids directly impact a number of proteins you identify in your work. If your CPAP cohort were prescribed glucocorticoids, you should mention this as a confounder. If they do not (perhaps because the samples were collected before the RECOVERY study released early results in June 2020), you should still mention this just to make clear to the reader that the medication regime should not be a major confounder beyond any treatments for comorbidities (also a potential confounder but less severe than dexamethasone being applied to a whole cohort and not the controls).
Line 106, for Table 1 it would be helpful to include p-values for the negative controls versus the total COVID-19 positive participants.
Line 170, MetaboAnalyst 4.0 should be cited as well as the web address provided
https://doi.org/10.1093/nar/gky310
Lines 171-173, it would aid reproducibility to briefly comment on any pre-processing of the data in MetaboAnalyst. You note that you have normalised to total peptide area – were the results investigated for heteroscedasticity, did you log transform, and were the data pareto or unit scaled, for example?
· Results
Lines 204-215, really this section is more of a Discussion than a simple reporting of Results; I would suggest that this paragraph be incorporated in the Discussion section
Figure 3a is interesting, but the simple 2D scatter plot of two components does not add much information without relevant R2Y or Q2Y values. Does separation improve at three or more components, measured by R2Y or Q2Y (or LOOCV accuracy)? This functionality is available within MetaboAnalyst. Also see comment in Discussion regarding noise in the dataset and overfitting.
· Discussion
The Discussion section is generally well-written and refers to appropriate literature.
As a reader, I was left a little confused about the small number of peptides in common. On an individual sample basis, what is the mean number of peptides identified per sample, and what is the standard deviation of this number? The reason for asking is to better understand whether the lack of overlap is due to biological variation and small n.
In your discussion of Table 2, it would be helpful to note the concordance between proteins you have identified in Table 2 as dysregulated in COVID-19 and those identified as modulated by glucocorticoid treatment, specifically APOC3, TTHY, ITIH2, HPT and C1QB (see reference below and supp materials thereof for full list of proteins). Suggest that you comment that several of the proteins identified in Table 2 overlap with those previously identified and support the hypothesised method of action of dexamethasone etc in treating COVID-19.
https://doi.org/10.3390/ijms232012079
You do not discuss Figure 3a. It is definitely odd that mild and severe separate, but that negative overlaps with both. To me this is suggestive of a noisy dataset and model overfitting. If your Q2Y values are low, and LOOCV accuracy shows poor classification, then this is suggestive of n being too small for multivariate classification. In other words, either method variation or biological variation is confounding separation analysis. With small n, you may wish to consider whether it is appropriate to include the PLS-DA analysis, whether you obtain better R2Y and Q2Y from a two-class separation (COVID positive versus negative), or whether to remove it given that your discussion does not review the PLS-DA at all.
Line 382, the word ‘it’ is missing. The text should read “However, it must be noted that …”
· Conclusions
Lines 406-408 essentially say the same as lines 401-402. Suggest removing 406-408 and rewording the earlier part as follows:
“Although to date this is the only study that investigated how SARS-CoV-2 infection affects circulating peptides, a larger cohort of patients, including the asymptomatic ones, would allow the identification of more specific pathways and peptides associated with the development of the disease. A larger study with greater statistical power would also help to overcome high inter-variability between subjects. A second limitation…”
Reviewer 2 Report
The authors make a try to profile free peptides in the plasma and relate their findings with severity in COVID-19 patients. Their work could be richer with more analyses included. Rather than that, they fall short in providing small and debatable changes in specific peptides. New knowledge provided by this paper is little. More specific comments:
- More Covid-related literature should be discussed in the introduction and maybe discussion. Particularly, Covid-proteomics studies are of high importance to be included and they are many.
- Why peptidomics and not proteomics? This should be discussed much more.
- Τhe use of free software (i.e. DIANN) in a library-free mode is recommended. Maybe more peptides will be detected.
- The cohort is small and unbalanced. Not sure if this can be accepted.
- Abundance boxplots rather than tables of fold-changes are strongly recommended?
- PLS-DA is a supervised method and problems of overfitting can occur. How about a PCA?
- More discussion and data about the peptides that are unique for each group should be included. If these data are not trustworthy for any reason, it should be discussed too.
- Fibrinogen can be highly affected by sample handling. Please discuss.
- Apolipoproteins should be examined with respect to any metabolic disorders in the cohort.
- TTR is related to feeding status. What is the feeding status of severe covid patients? Does this affect TTR abundance?
- line 281: "were reported to be unaffected by COVID-19". literature?
- line 389: "abundances are strongly correlated". the word strongly should be omitted
Round 2
Reviewer 2 Report
The authors replied to all comments. For some of them concerning literature and some plots asked (PCA, boxplots), they answered and fulfilled the demands.
Nevertheless, there are some critical points that the answers did not cover what was required. Particularly, their responses no 7 and 9 reveal inconsistencies of the provided data. This occurs mostly because of the limited number of samples as the authors agree.
Regarding fibrinogen we have to accept the authors' reassurance of consistent sampling, but we cannot be satisfied with answers about TTR and apolipoproteins, since they are among the important findings of the study.
